# Biodegradation of Polyurethane by Fungi Isolated from Industrial Wastewater—A Sustainable Approach to Plastic Waste Management

**DOI:** 10.3390/polym16101411

**Published:** 2024-05-16

**Authors:** Aiswarya Rajan, Fuad Ameen, Ranjitha Jambulingam, Vijayalakshmi Shankar

**Affiliations:** 1School of Biosciences and Technology, Vellore Institute of Technology, Vellore 632014, India; 2Department of Botany and Microbiology, College of Science, King Saud University, Riyadh 11451, Saudi Arabia; 3CO_2_ Research and Green Technologies Center, Vellore Institute of Technology, Vellore 632014, India

**Keywords:** polyurethane, biodegradation, optimization, analysis

## Abstract

Polyurethane (PU) is a type of polymer, which exists in various forms in the environment. Very few studies are available concerning the structure or enzymatic mechanism of the microbial community, which can degrade PU. Degradation of PU remains a difficult problem with respect to the environmental and biological disciplines. This study mainly focused on identifying the micro-organisms able to degrade polyurethane and confirming the degradation by performing a plate assay, Sturm test and scanning electron microscopy. Optimal culture conditions for maximum PU degradation were also analyzed through classical methods. A soil burial test was conducted by placing polyurethane films in the soil for one month, and the microbe growing on the surface of polyurethane films—with a maximum degradation of 55%—was isolated and identified as *Aspergillus versicolor* (ARF5). The culture medium was also optimized with different physical and chemical parameters for maximum PU degradation. The presence of CO_2_ as a by-product of PU biodegradation was confirmed through the Sturm test.

## 1. Introduction

Plastic is a generic term representing the polymer groups, which is used extensively in day-to-day life by industries due to its simple manufacturing process, lucrative and superior properties and high resistance to degradation [1]. Without proper management, plastic waste may cause serious damage to the environment and living organisms [2]. Otto Bayer in the year 1937 developed the first polyurethane polymer, and later, it was used in industries [3]. Polyurethane was recently synthesized by reacting a chain extender with polyol in the presence of di-isocyanate [4]. Based on the polyol type, polyurethane is categorized into four different types, namely polyacrylic, polycaprolactone, polyether and polyester urethanes. Among the four types of polyurethanes, polyester polyurethanes are the most durable and recalcitrant ones compared to other types [5]. 

Polyurethanes are a significant and also adaptable class of man-made polymers utilized in a wide range of medical, automotive, as well as industrial products. Polyurethane is obtained as a result of a chemical reaction between poly isocyanates and polyalcohol. Polyol is mostly composed of polyester or polyether [6]. The building blocks associated with the synthesis of polyurethane control the final macromolecular design, crystallization and molar mass, which are high parameters, affecting the biodegradability of polyurethane. For example, polyester-based polyurethane is known to be more biodegradable than polyether due to ester bonds, which are hydrolyzed [7]. 

The accessibility of polyurethanes to degrading enzyme systems is based on several polymer features, including the molecular orientation, crystallinity, cross-linking and chemical groups contained in the molecular chains [8,9]. The majority of plastic deterioration is caused by micro-organisms, whereas abiotic processes, such as hydrolysis and UV degradation, have a relatively small impact. Fungal degradation, bacterial degradation and degradation by enzymes are the three main kinds of polyurethane degradation [10,11,12,13,14,15]. The significant and adaptable class of synthetic polymers, known as polyurethanes, is utilized in a wide range of industrial, automotive and medicinal products [16,17,18].

Polyurethane biodegradation is highly dependent on fungal, bacterial and enzymatic degradation [19,20]. Polyester-based polyurethanes are more vulnerable to fungal attacks. Fungal communities in the soil are associated with degradation of polyurethane [19]. For example, *Curvularia senegalensis*, *Fusarium solani*, *Aureobasidium pullulans* and *Cladosporium* were isolated from soil and found to degrade from ester-based PU [20]. In the intracellular degradation of polyurethane, extracellular PU esterase and PU esterase play important roles. Intracellular access to the hydrophobic polyurethane surface is given by the enzyme [21,22,23]. At this point, the enzyme adheres to the polyurethane surface. The enzyme allows the hydrophobic polyurethane surface to be accessed from inside the cell. 

The waste generated by PU degradation is mainly disposed of in municipal landfills. Due to its structural complexity, it may produce hazardous substances, which are harmful to the environment and human health. PU biodegradation remains a challenge for the environmental and biological disciplines, as very little is known regarding its structure or the degradative enzymatic pathway of the microbial community, which is capable of PU degradation [24,25,26,27]. Few research works have been published previously analyzing the biodegradation of PU and most of them were concentrating on bacterial enzymes degradation. There are still difficulties associated with the biological breakdown of PU. Biodegradation of polyurethane with fungi has not been explored much. We attempted to optimize polyurethane degradation with *A. versicolor* by varying different parameters. This study mainly focused on polyurethane degradation by *Aspergillus versicolor* isolated from the soil sample and optimized the best culture conditions for maximum degradation. 

## 2. Materials and Methods

### 2.1. Chemicals and Reagents

Polyurethane (polyester) beads and tetrahydrofuran (Figure 1a) were purchased from Sigma-Aldrich, St. Louis, MO, USA. The composition of the mineral salt medium (MSM) was as follows: (g/L: dipotassium phosphate (K_2_HPO_4_) 0.5, mono-potassium phosphate (KH_2_PO_4_) 0.04, sodium chloride (NaCl) 0.1, calcium chloride dehydrate (CaCl_2_H_4_O_2_) 0.002, ammonium sulfate ((NH_4_)_2_SO_4_) 0.2, magnesium sulfate heptahydrate (MgSO_4_·7H_2_O) 0.02, ferrous sulfate (FeSO₄·H₂O) 0.001, pH adjusted to 7.0 by adding NaOH/H_2_SO_4_), nutrient agar, Sabouraud dextrose agar from Sigma-Aldrich and glucose (Figure 1). The remaining reagents were commercially accessible products of the highest quality [6,28]. 

### 2.2. Preparation of Thin Polyurethane Films

Approximately 1 g of polyurethane pellets was placed in 100 mL of tetrahydrofuran in a 250 mL beaker and sonicated for 30 min for complete dissolution. This mixture was added to four clean-glass Petri dishes in equal quantities to prepare the PU thin films. By placing the covered Petri dishes in a desiccator for 48 h, tetrahydrofuran was able to slowly evaporate. After a period of 48 h, structural changes in the polymer were analyzed, and the film was removed. Then, the film was washed with sterilized distilled water and stored at room temperature (Figure 1b–d) [28,29]. 

### 2.3. Sample Collection

Industrial wastewater was collected from the plastic processing industry in Kerala kozhikode. Wastewater samples were collected in sterile polyethene bags. Large particles (wood bits, plastic and paper bits) were removed from the water samples by sieving. The collected samples were stored at 4 °C before processing.

### 2.4. Soil Burial Test

The prepared PU films were washed with sterilized distilled water and buried for six months in the soil containing mineral salt solution (g/L: dipotassium phosphate 0.5 g, monopotassium phosphate 0.04 g, sodium chloride 0.1 g, calcium chloride dehydrate 0.002 g, ammonium sulfate 0.2 g, magnesium sulfate heptahydrate 0.02 g, ferrous sulfate 0.001 g), and 2 g glucose was mixed in this solution [28]. The MSM solution was sterilized by autoclaving for 1 h and, after cooling, the solution was added to the soil pot. After 1 h, the PU film was removed from the soil pot, and sterilized distilled water was used to wash it away. The PU films were shifted into Sabouraud dextrose agar containing Petri plate and incubated at 36 °C for one week.

### 2.5. Isolation of Selected Micro-Organism

For the isolation process, the nutrient agar (NA) and Sabouraud dextrose agar (SDA) were prepared. PU films were shifted into NA and SDA plates for bacterial and fungal colony isolation, respectively; the media were sterilized further by autoclaving at 126 °C for 1 h. After sterilization, the sample was inoculated on NA and SDA using the streak plate method and the spread plate method and incubated for 48 h. Further, the selected microbes were used for analysis.

### 2.6. Screening and Identification of Selected Colonies 

Selected colonies with strong zones of hydrolysis around them due to PU degradation were chosen for further examination. The selected microbe was identified based on its morphological structure. Then, the identified microbes were stored until further study [28,29,30].

### 2.7. Determination of Growth and PU Degradation

In order to determine the growth curve of fungi, cell dry biomass weight and PU degradation were calculated. The media were incubated at 37 °C at different intervals (24 h, 48 h, 72 h, 96 h, 120 h, 144 h and 168 h). The sample was withdrawn and checked for biomass concentration and degradation of PU.

### 2.8. Degradation Analysis

#### 2.8.1. Screening of Polyurethane Degrading Micro-Organisms Using the Clear Zone Method

Polymer degradation ability was initially screened using the clear zone test. Approximately three polyurethane films were added to the mineral salt medium at a final concentration of 0.1% (*w*/*v*) and sonicated for 1 h at 120 rpm in a shaker. After sonication, the medium was sterilized at 121 °C and pressurized at 15 psi for 20 min. Approximately 15 mL of the sterile medium was poured into each plate before cooling [15]. The agar-plate-containing polymer was inoculated with the selected colony and incubated at 37 °C for 3 days. The plates were filled with 0.1% (*w*/*v*) solution of Coomassie Brilliant Blue R 250 in 10% (*v*/*v*) acetic acid with 40% (*v*/*v*) methanol for 20 min, and the dye was poured; then, the method was repeated using methanol and acetic acid. Areas of clear degradation were visualized with a blue background [28]. 

#### 2.8.2. SEM and FTIR Analyses

The test samples and the control samples were analyzed using scanning electron microscopy and Fourier transform infrared spectroscopy (SEM and FTIR). For SEM analysis, the treated and untreated samples were thoroughly washed with double-distilled sterile water. After drying, the samples were mounted by coating with silver in vacuum on aluminum stubs. Both the treated and untreated sample images were compared. For the analysis of samples using FTIR (FT/IR-6X FTIR spectrometer, Jasco, Tokyo, Japan), PU films were mixed with KBr after degradation to form a pellet and mounted on the sample plate. A spectrum was measured from the wavenumbers 400–4000 cm^−1^ to determine the functional group changes in the treated and untreated samples. 

#### 2.8.3. Sturm Test

CO_2_ evolution as a result of PU biodegradation was analyzed through the Sturm assay [29]. The selected isolate was used as an inoculum. Two flasks were used for the test, and the flask containing 3 gm of polyurethane pieces was used as a substrate and mixed with the inoculum (5%) in MSM, whereas the control flask contained the inoculum in MSM medium without polyurethane pieces and was kept at room temperature for four weeks with continuous stirring. The amount of CO_2_ generated and the change in biomass (CFU/mL) in the test flask and gravimetric control were determined. The evolution of CO_2_ as a result of polymeric chain degradation was trapped in the absorption flasks containing 1 M KOH. In the KOH flask containing CO_2_, 0.1 M BaCl_2_ solution was added, and as a result, BaCO_3_ precipitates started to develop. CO_2_ production was determined gravimetrically by estimating the sum (g) of carbon dioxide precipitates via the addition of barium chloride [6,28].

### 2.9. Optimization of Media for Maximum PU Degradation

Various environmental factors—such as the temperature, pH, aeration and addition of nutrients to the medium—play a major role in polyurethane degradation, as they affect the activity of the microbe in the degradation process. Hence, it is important to optimize the culture media for maximum polyurethane degradation.

#### 2.9.1. Effect of Physical Parameters on PU Degradation

##### Effect of Temperature

Temperature is considered to be an important key factor for the degradation of polyurethane, as it affects the growth and activity of the microbe [31]. Polyurethane degradation was measured at different temperatures (25–55 °C) to determine the optimum temperature for maximum degradation. The culture was inoculated in polyurethane containing Sabouraud dextrose broth at pH 6.5 and incubated at different temperature ranges—from 25 to 55 °C—in an incubator. After 16 days of incubation, the degradation percentage was measured in each flask. 

##### Effect of Aeration

In order to determine the effect of aeration/agitation rate on polyurethane degradation, different agitation rates were maintained, and the flask in the static condition was used as control. The culture was inoculated in the polyurethane film containing Sabouraud dextrose broth (pH 6.5) and incubated at different agitation speed ranges—from 50 to 200 g-force—in an orbital shaker. After 16 days of incubation, the degradation percentage in each flask was measured. 

##### Effect of pH

Polyurethane degradation was measured at different pH levels—from 5 to 7—to determine the optimum pH condition for maximum degradation. The sample was prepared with Sabouraud dextrose broth by adjusting the pH at different levels (5–7) and incubated at 37 °C in an incubator. After 16 days of incubation, the degradation percentage was measured. The maximum polyurethane degradation was found to be correlated with fungal growth.

#### 2.9.2. Effect of Nutrients on PU Degradation

Different carbon (mannitol, glucose, starch) and nitrogen sources (ammonium sulphate, urea and calcium nitrate tetrahydrate) were added to the medium to check its effect on polyurethane degradation. After 16 days of incubation, the degradation percentage was measured. 

## 3. Results and Discussion 

### 3.1. Isolation of PU Degrading Strains

The prepared polyurethane films were buried in the soil for a period of one month; afterward, the buried films were taken out, washed and used to isolate the PU degrading micro-organisms. In total, twelve different types of microbes were isolated from the polyester pieces after being buried in soil for 1 month (Figure 2). Each isolated microbe was subjected to screening for polyurethane degradation. Some studies documented polyurethane degradation using fungi and bacteria [31,32], such as *Penicillium, Aspergillus, Pullularia, Trichoderma* and *Chaetomium*, by breaking the ester bond in polyurethane; however, a complete breakdown of polyurethane still remains a challenge. Some microbes, such as *Corynebacterium* species, *Enterobacter agglomerans* and *Bacillus subtilis*, were found to degrade polyurethane samples effectively [6,28,33,34,35].

Compared to other polyurethane forms, polyester polyurethane was found to be more vulnerable to attacks by microbes due to the presence of linkages between urethane and the ester bond. Filamentous fungi have the ability to form colonies and have the potential to secrete a variety of enzymes, which will easily degrade the different organic substances. Of the isolated microbes, some had the capacity to degrade polyurethane films.

### 3.2. Screening of Micro-Organisms 

In general, degradation ability is based on the growth of the microbe; hence, the screening part is highly crucial. The serial dilution method was used in this study to estimate the concentration of soil samples (number of colonies) by counting the number of colonies obtained from serial dilutions of the samples. The isolated microbes are tabulated in Table 1. Of the twelve species, one microbe (ARF5) showed the highest degradation rate, and hence, it was chosen for further studies and identification. 

The isolated culture was grown for 28 days at 25 °C in potato dextrose agar. An interference contrast microscope was used to investigate the fungus’ morphology. For molecular identification, modified versions of the methods employed by Hong et al. were utilized. Random amplified polymorphic DNA–polymerase chain reaction (RAPDPCR) was conducted using universal fungal primers. The DNA was isolated from the fungi sample, and it was amplified. A comparison of the amplified sequence with the sequence available in GenBank was performed, and the results showed a high sequence similarity (98%) with *Aspergillus versicolor*. The sequence was submitted to GenBank, and the accession number is OL614816. 

The isolated fungi were identified as *Aspergillus versicolor* (ARF5), which showed the highest degradation percentage of polyurethane films. Of the twelve other species isolated, other polyurethane degrading microbes, such as *Aspergillus niger*, *Fusarium solani, Cladosporium* sp., *Comamonas acidovorans, Pseudomonas fluorescens chlororaphis* and *Pseudomonas chlororaphis*, were also isolated from garbage soil samples and proceeded to degrade ester-based PU. The colony morphology of the isolated organism (*Aspergillus versicolor)* had a spherical shape, with long, smooth, colorless conidiophores and Biseriate; phialides and round, loosely radiated head vesicles were found. *Aspergillus versicolor* (ARF5) was selected for further study (Table 2).

### 3.3. Optimization Parameters for Polyurethane Degradation 

#### 3.3.1. Effect of Temperature 

The PU degradation ability of *Aspergillus versicolor* was tested at different temperatures from 25 °C to 55 °C. It was found that maximum growth and degradation of polyurethane (49%) was observed at 35 °C, with a degradation of 500 mg/L of PU after 16 days of incubation. Based on these results, it was identified that polyurethane degradation is highly correlated with the growth of fungi, as well as incubation temperature. The higher the degradation rate, the higher the growth rate of fungi. At a higher temperature (55 °C), the growth as well as degradation were found to be lower (33%) (Figure 3). This suggests that a faster cell growth rate will lead to an increased rate of degradation. Hence, it was found that maximum PU degradation was observed when cultivating *Aspergillus versicolor* at 35 °C. A study by Urooj Zafar et al., 2013 [36], showed polyurethane degradation at different temperatures, such as 25, 45 and 50 °C, and identified that *Fusarium solani* and *Candida ethanolica* played a major role in the degradation process. Khan et al., 2020 [37], discovered that the ideal temperature for *A. flavus G10* was 25 °C, and the rate of biodegradation decreased with the increase in temperature. Khan et al. [6] previously established significant esterase activity in polyurethane degradation in *A. tubingensis* liquid broth culture incubated at 37 °C.

#### 3.3.2. Effect of Aeration on Polyurethane Degradation 

The influence of aeration on polyurethane degradation was tested by incubating the microbe at different aeration conditions (50–200 rpm) in an orbital shaker. The degradation rate and microbial activity were greatly affected by the aeration rate [38]. Maximum growth and polyurethane degradation by *Aspergillus versicolor* were identified at 150 rpm. A constant increase in growth and degradation was observed with the agitation speed from 50 to 150 rpm, with an increase in degradation rate percentage from 20 to 55% (Figure 4). While the culture was incubated in a static condition, a very minimal degradation rate was observed. Based on the literature, it was identified that a maximum degradation percentage was observed under higher agitation. Hence, we can conclude that an increase in agitation speed will increase the cell mass, which will result in a better degradation capability.

#### 3.3.3. Effect of pH

In order to determine the optimum pH condition for maximum growth and polyurethane degradation, the culture medium was incubated at different pH ranges from 5 to 7 for 16 days. Degradation is greatly influenced by pH changes in the growth medium, as it plays an important role in the growth of the microbe; hence, it is highly essential to optimize the pH condition. *Aspergillus versicolor* growth and polyurethane degradation were significantly influenced by the initial pH of the culture. At pH 5, although growth was not completely inhibited, PU degradation was found to be very minimal. Maximum polyurethane degradation was observed at pH 6 up to 16 days (Figure 5). Similarly, Majid H. Al-Jailawi et al., 2015 [39], observed maximum growth and polyethylene degradation by *P. putida* at pH 6.5. In contrast, Islami et al., 2019 [40], showed higher LDPE degradation by *Clostridium* sp. at pH 7.0. Another study showed that the most appropriate pH for PU breakdown by *A. tubingensis* was acidic pH [6].

#### 3.3.4. Effect of Nitrogen Sources 

The impact of utilizing different nitrogen sources (1%) on the growth of *Aspergillus versicolor* and its PU degradation was tested with three different nitrogen sources. Compared to urea and calcium nitrate, ammonium sulfate was found to enhance fungal growth and polyurethane degradation. Although urea stimulated fungal growth, PU degradation was suppressed after 16 days of incubation (Figure 6). When culturing with calcium nitrate tetrahydrate, the growth and degradation were slightly improved. It is evident that, compared to other nitrogen sources, the presence of ammonium sulfate in the culture medium resulted in a high degradation percentage; hence, it was chosen as an optimum source for maximum polyurethane degradation. A study by Toshiaki Nakajima-Kambe et al., 1995 [33], reported that *Comamonas acidorvorans* utilized polyurethane as the sole source of nitrogen, resulting in degradation of polyester polyurethanes. 

#### 3.3.5. Effect of Carbon Source

The impact of utilizing different carbon sources (1%) on the growth of Aspergillus versicolor and its PU degradation was analyzed by adding three different carbon sources, namely glucose, mannitol and starch, in the culture medium. Aspergillus versicolor was grown in Sabouraud broth in the presence of three different carbon sources (Figure 7). It was observed that mannitol led to an increase in fungal growth and PU degradation. Meanwhile, glucose promoted slight fungal growth and degradation over 16 days of incubation. A report by Oceguera Cervantes et al., 2007 [41], showed that *Alicycliphillus* sp. was found to utilize polyurethane as the sole source of carbon. 

### 3.4. Determination of Growth and PU Degradation

In order to determine the growth curve of fungi, cell dry biomass weight and PU degradation were calculated at different intervals (24 h, 48 h, 72 h, 96 h, 120 h, 144 h and 168 h). Maximum growth was obtained on day 7. After 5 days of growth, 15.5 g/L of fungal dry biomass was obtained, and polyurethane degradation was identified at 36%. Growth and degradation followed the same pattern (Figure 8).

### 3.5. Degradation Studies

In this study, polyurethane degrading ability was identified using the plate assay method. After incubation, a clear zone due to hydrolysis was observed around the colonies, which indicated the production of the polyurethanase enzyme. This test determines the depolymerization ability of the substrate by the enzyme when it interacts with Coomassie blue R 250, which results in formation of a zone of clearance with blue background via depletion of the polymer material. Clear regions of inhibition of 13 mm around the selected fungal colony were observed in mineral salt agar plates containing Coomassie blue R 250. 

In this work, the Sturm test was used for the analysis of polyurethane biodegradation in a liquid medium. Carbon dioxide release indirectly represents the metabolism and growth rate of the microbe on the polyurethane film. The total quantity of CO_2_ produced was 4.48 g/L, while without polyurethane, the control amount was 2.24 g/L (Table 3). Compared with the control reaction vessel, the increase in CFU/mL and the amount of CO_2_ developed within the test indicated an increase in the activity of fungi on polyurethane films. Some researchers have used the Sturm analysis to investigate how biodegradable polymers degrade in aliphatic as well as aromatic compounds [40,42]. This result was in accordance with another report by Shah et al., 2013, which investigated the amounts of CO_2_ released through the activity of *P. aeruginosa* and *B. subtilis* on polyurethane films, which were found to be 6.54 and 7.08 g/L, respectively [43]. Several studies have conducted carbon dioxide release tests in order to determine the polymer degradability of microbes, which were found to release higher quantities of carbon dioxide when cultured with a polymer as the sole source of carbon [39,40].

The changes in surface morphology of polyurethane films were identified using SEM and FTIR analyses. PU films procured commercially were observed as films and foams [42]. The result was similar to a study by Álvarez-Barragán et al., 2016 [44], who also observed the presence of holes in the films after microbial treatment. In order to determine the degradation of polyurethane films, microbial growth on the surface of films was investigated using SEM. Both the control and test films were analyzed using scanning electron microscopy, and it was observed that films incubated with sterile media showed an absence of microbes, and the test film showed microbial colonies on the films (Figure 9). When analyzed under scanning electron microscopy, the test films showed visual signs of degradation, such as rough surface with numerous holes, pits, bends, fracturing, surface crumbling, fissure formation, pores and cracks, due to microbial activity on the films, while the control film was found to be clear, with a smooth surface (Figure 10). With these results, it is evident that the presence of microbial activity on PU films resulted in morphological changes in the films and polyurethane degradation.

The polyurethane film after microbial treatment was analyzed using FTIR spectroscopy (Figure 11) to determine the structural changes in functional group representatives, such as amide, urethane and ester [45]. It was observed that compared to the control film without microbial treatment, the microbial treated film showed various functional groups at different intensities. Figure 1b indicates the appearance of some new peaks (shown as C=C) and an increase in already existing peaks—in the region of 1400–1600 cm^−1^—indicating the formation of new intermediate products (1440 cm^−1^). The bonds are related to strong bands in the range of 1300–1000 cm^−1^ (asymmetrical and symmetrical O–C–O stretching vibrations). Similar to this result, Natasha R. Gunawan et al., 2022 [46], analyzed the degradation of polyester polyurethane by a marine microbe and observed a peak at 1220 cm^−1^, indicating C–O stretching. The range of 1750–1720 cm^−1^ (1724 cm^−1^) indicates the strong ester bonds’ (C=O) breakage of ester moieties from the backbone of polyurethane chain, which leads to the synthesis of smaller compounds, such as primary amine ends and hydroxyls. This shows that the polyurethane films were depolymerized with the tested microbe. With these FTIR results, we can conclude that the polyurethane film can be degraded with an isolated microbe. 

## 4. Conclusions

Polyurethane has been widely used in different manufacturing processes, such as paints, footwear, furniture and medical devices, and its degradation is highly important, as its usage is increasing day by day. Biodegradation of polyurethane (PU) has been poorly explored, and most of the published studies have focused on bacterial enzymes. PU biodegradation is still a challenge for the environmental and biological disciplines, as very little is known regarding its structure or the degradative enzymatic pathway of the microbial community, which is capable of PU degradation. The majority of plastic degradation is caused by micro-organisms, whereas abiotic processes, such as photodegradation and hydrolysis, have a relatively minimal impact [1,32,34,35,36,37,38,39,40]. In this work, fungi were used to degrade PU. It appears that the biological degradation of PU remains a challenge. Upon a preliminary screening of PU degrading fungi on an SDA plate, many bacterial colonies were observed, but we limited our work to fungi and re-cultured the specific fungi only. An optimization of the culture conditions for maximum polyurethane degradation was calculated through classical methods. In this study, degradation was confirmed through various instrumental analyses, such as FTIR, SEM and the Sturm test. It was demonstrated that *Aspergillus versicolor* had a higher ability to degrade PU compared to other microbes, which were isolated from the garbage soil. The optimum culture conditions for maximum polyurethane degradation were pH 6 and temperature 35 °C. Finally, it was revealed that *A. versicolor* was related to in situ biodegradation of PU.

## Figures and Tables

**Figure 1 polymers-16-01411-f001:**
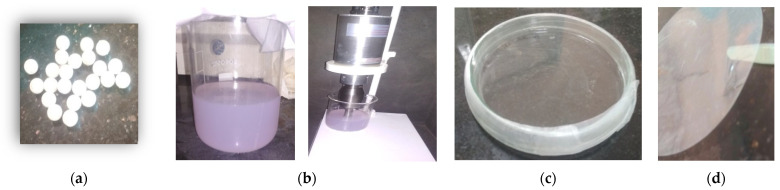
(**a**) Polyurethane beads; (**b**) THF solution blended with PU beads; (**c**) THF solution in sonicator; (**d**) Polyurethane film.

**Figure 2 polymers-16-01411-f002:**
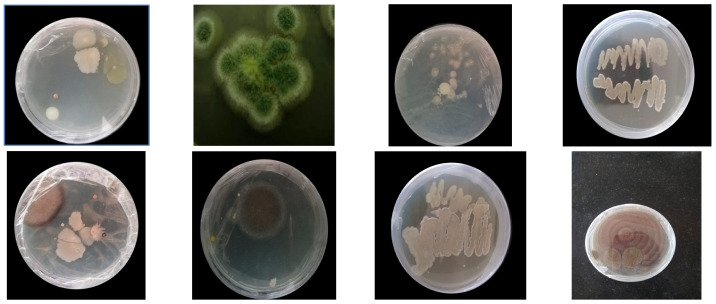
Isolated cultures.

**Figure 3 polymers-16-01411-f003:**
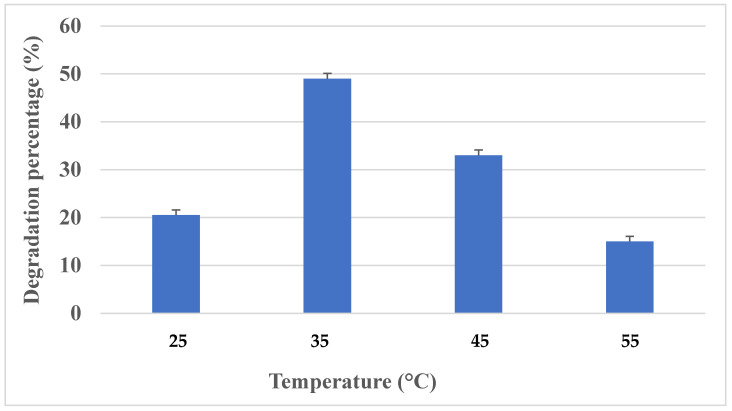
Effect of temperature.

**Figure 4 polymers-16-01411-f004:**
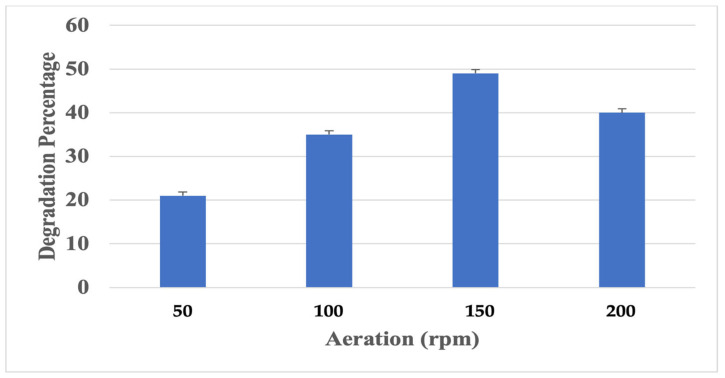
Effect of aeration.

**Figure 5 polymers-16-01411-f005:**
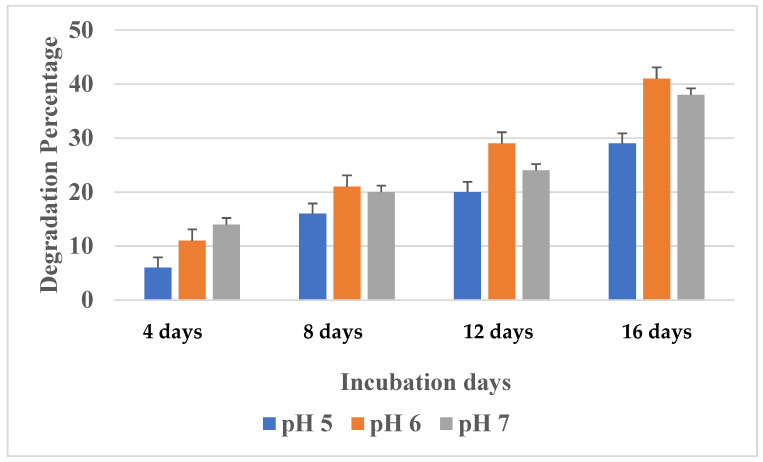
Effects of pH on PU degradation.

**Figure 6 polymers-16-01411-f006:**
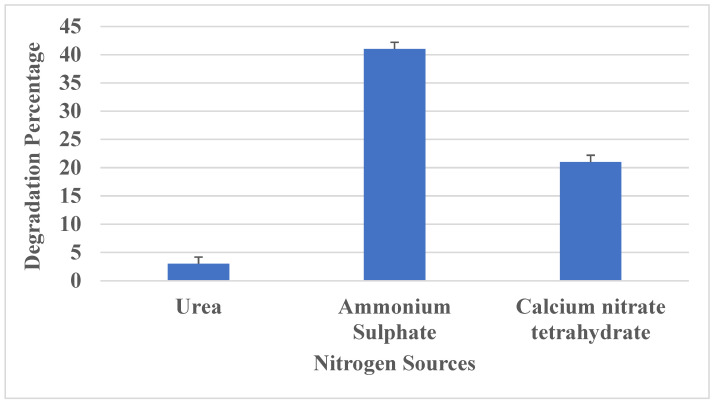
The effect of various nitrogen sources on PU degradation.

**Figure 7 polymers-16-01411-f007:**
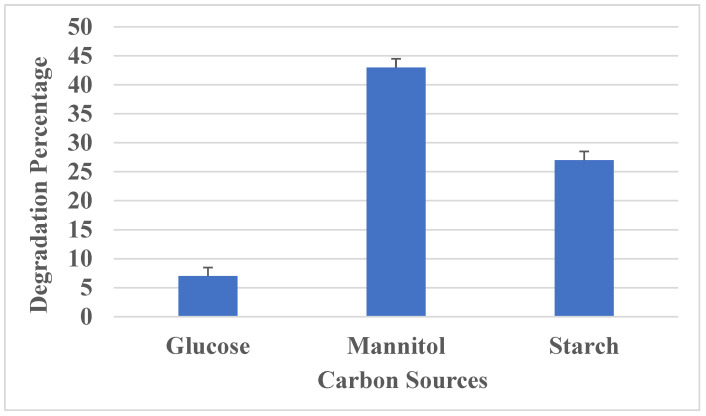
Effect of carbon sources on PU degradation.

**Figure 8 polymers-16-01411-f008:**
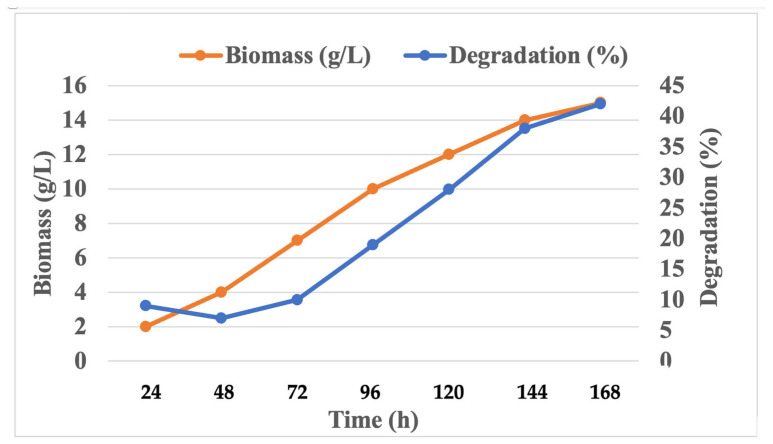
Growth curve and degradation (%).

**Figure 9 polymers-16-01411-f009:**
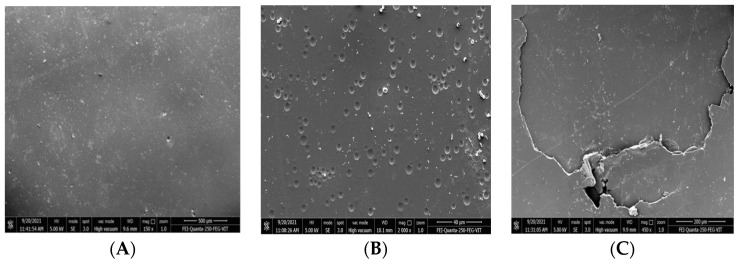
(**A**) Before degradation; (**B**,**C**) After degradation.

**Figure 10 polymers-16-01411-f010:**
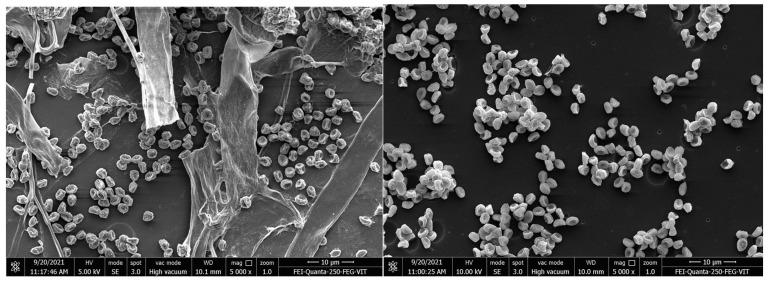
Microbial growth on a polyurethane film.

**Figure 11 polymers-16-01411-f011:**
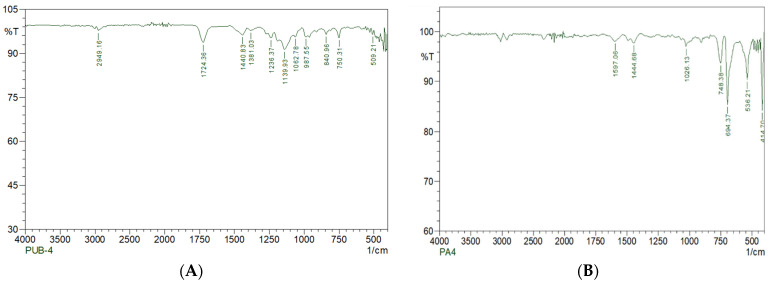
(**A**) Before degradation; (**B**) After degradation.

**Table 1 polymers-16-01411-t001:** Isolation of micro-organisms from garbage soil growing in the presence of polyurethane films.

S. No	Isolated Micro-Organism	Type of Organism	Colony Color	Genus Name as per Morphology	Degradation Percentage (A_o_ − A_t_/A_o_) × 100
1	ARF1	Fungi	Dark green	*Aspergillus* sp.	40%
2	ARF2	Fungi	Orange	*Penicillium* sp.	16%
3	ARF3	Fungi	Gray	*Botrytis* sp.	55%
4	ARF4	Fungi	White	*Candida* sp.	17%
5	ARF5	Fungi	Green	*A. versicolor*	58%
6	ARF6	Fungi	Black	*A. niger*	19%
7	ARB2	Bacteria	Pink	*Serratia* sp.	3%
8	ARB3	Bacteria	Red	*Serratia* sp.	28%
9	ARB4	Bacteria	White	*Bacillus* sp.	12%
10	ARB5	Bacteria	Pale yellow	*Staphylococcus* sp.	24%
11	ARB6	Bacteria	Orange	*Rhodotorula* sp.	17%
12	ARB7	Bacteria	Green	*Pseudomonas* sp.	19%

Where A_o_ is the absorbance at time t = 0 min, and A_t_ is the absorbance after treatment time.

**Table 2 polymers-16-01411-t002:** Characteristics of *Aspergillus versicolor* ARF5.

Characteristics	*Aspergillus versicolor* ARF5
Shape	Spherical
Stipes colour	Yellow
Surface	cheese surfaces
Conidia surface	Slightly rough
Metula covering	1/3 to entire vesicle
Vesicle serration	Biseriate

**Table 3 polymers-16-01411-t003:** Sturm test calculating the total viable count and gravimetric analysis of CO_2_ evolution through polyurethane breakdown.

Case Study	Before the Experiment (CFU/mL)	After the Experiment (CFU/mL)	CO_2_ Emitted per Liter (g/L)
Test	4.6 × 10^9^	9.8 × 10^9^	4.48
Control	4.6 × 10^9^	2.3 × 10^9^	2.24

## Data Availability

All data related to this manuscript are incorporated in the manuscript only.

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
