# Peer review of "Biodegradation of Polyurethane by Fungi Isolated from Industrial Wastewater—A Sustainable Approach to Plastic Waste Management"

_polymers, 2024, doi:10.3390/polym16101411_

Round 1
Reviewer 1 Report (Previous Reviewer 2)
Comments and Suggestions for Authors
No more suggestions
Author Response
We thank the reviewer for accepting the manuscript
Reviewer 2 Report (Previous Reviewer 1)
Comments and Suggestions for Authors
The manuscript entitled “Biodegradation of Polyurethane by fungi isolated from Industrial wastewater – A sustainable approach for plastic waste management is devoted to an important problem searching for biotechnologies for cleaning the environment from polymer pollutants.
This is not the first time I have reviewed this manuscript, and compared to the last time it has become noticeably better. And in general it corresponds to the main scopes of “Polymers” journal.
There are a number of comments regarding the text.
1. The text lacks fundamental and applied components. Authors should describe the taxonomic characteristics of the isolated strains rather than simply describing the color of the colonies. If the authors discovered changes in functional groups on the surface of the polymer, it is worth connecting with what biochemical processes they can change. Polymers Journal is in Materials Science field, you need more connections with the structure of polyurethane and the biochemical features of its changes. It was worth taking a closer look at the technological solutions for using the selected strains.
2. I have questions about the composition of the cultivation medium. It seems to me that there is an error in the concentration of a number of components; they are too small.
3. Descriptions of devices must be given separately.
4. in Keralako zhikode. Map, coordinates...
5. How was the degradation percentage calculated? It is usually calculated by the change in mass of the material during the degradation process. I didn't see any descriptions in the methods.
6. 210-214 makes sense if the taxonomic positions of the isolates are given. In general, to provide photos of isolates, not comme il faut, taxonomic characteristics are needed. They are not expensive or difficult to make.
7. Quality of figures and signatures. The Excel illustrations resemble student work. Figure 8 9 with three columns is not very informative.
8. In conclusion, it is worth paying attention to practical use.
9. Authors should focus on the scientific novelty of the manuscript (or at least practical) until it is obvious.
Comments on the Quality of English LanguageModerate editing of English language required (style)
Author Response
The manuscript entitled “Biodegradation of Polyurethane by fungi isolated from Industrial wastewater – A sustainable approach for plastic waste management is devoted to an important problem searching for biotechnologies for cleaning the environment from polymer pollutants.
This is not the first time I have reviewed this manuscript, and compared to the last time it has become noticeably better. And in general it corresponds to the main scopes of “Polymers” journal.
There are a number of comments regarding the text.
- The text lacks fundamental and applied components. Authors should describe the taxonomic characteristics of the isolated strains rather than simply describing the color of the colonies.
Response: We thank the reviewer for the comments. As per the suggestions, we have included the taxonomic characteristics of the isolated microbe based on the morphology in the table 1.
- If the authors discovered changes in functional groups on the surface of the polymer, it is worth connecting with what biochemical processes they can change. Polymers Journal is in Materials Science field, you need more connections with the structure of polyurethane and the biochemical features of its changes. It was worth taking a closer look at the technological solutions for using the selected strains.
Response: We thank the reviewer for the suggestion. In the section 3.5, FTIR results has been elaborated about the changes in the functional groups before and after degradation.
- I have questions about the composition of the cultivation medium. It seems to me that there is an error in the concentration of a number of components; they are too small.
Response: We thank the reviewer for the comments. We have chosen important parameters which affect the degradation. Totally five different parameters have been chosen and studied in detail.
- How was the degradation percentage calculated? It is usually calculated by the change in mass of the material during the degradation process. I didn't see any descriptions in the methods.
Response: We thank the reviewer for the comments. In the table 1 the formula to calculate the degradation percentage was given.
- Quality of figures and signatures. The Excel illustrations resemble student work. Figure 8 9 with three columns is not very informative.
Response: We thank the reviewer for the suggestions, we have improved the quality of the figure.
- In conclusion, it is worth paying attention to practical use.
Response: We thank the reviewer for the suggestions, we have considerably improved the conclusion section.
- Authors should focus on the scientific novelty of the manuscript (or at least practical) until it is obvious.
Response: We thank the reviewer for the comments, the novelty of the study was mentioned at the end of the introduction section.
Round 2
Reviewer 2 Report (Previous Reviewer 1)
Comments and Suggestions for Authors
After the corrections, the manuscript became much better. I advise authors to correct the figure captions. They must be expanded and must show the meaning of the drawing, allowing it to be understood without the text of the manuscript. There are no error bars in Figure 10. The design of the figures does not meet the quality requirements of the magazine. Authors should download several issues and see how to design drawings and captions for them.
Comments on the Quality of English Languageauthors should correct the style of the text in some places, using the services of a native speaker
This manuscript is a resubmission of an earlier submission. The following is a list of the peer review reports and author responses from that submission.
Round 1
Reviewer 1 Report
Comments and Suggestions for Authors
This is my second time reviewing this manuscript. Compared to last time, there are some improvements in the quality of the material. In general, this article corresponds to the goals and objectives of the journal Polymers. However, it needs serious improvement. Firstly, the style of the text should be seriously revised until the impression remains that the article was written by students and was not checked by the professor. A significant improvement in the scientific style of the text is needed. Second, the manuscript requires proofreading by a native English speaker. A number of sentences are not written clearly. thirdly, a significant improvement is required in the quality of the text and illustrations (captions under the figures), the quality of the diagrams, many illustrations are auxiliary, which can be cited when defending a diploma or course work, but not in a scientific article (Figure 2, 4, etc.). All illustrations with three or four columns should be combined into one. illustrations should be done in a more specialized editor. Excel is not comme il faut. It is completely unclear with which organisms the experiments were carried out; the authors isolated many strains, but the data are not clear on which ones are given. The title of the article contains fungi, but they also worked with bacteria, and the authors all call microbes. a table is needed with all the isolated strains and their identification, the strains need to be numbered and all results must be given with these numbers. The introduction resembles a student's thesis in style and materials. For example, the phrase We found very few literatures that deals with biodegradation with fungi. In my opinion, this indicates a problem with authors, but not a lack of literature.
The authors identified the strains; it is necessary to find more specific data on these strains in the literature and present them to formulate the goal of the work.
Thus, I cannot recommend this manuscript for publication in this form, due to its very low quality (I have not listed half of the obvious problems), but it has potential for publication. It is worth finding a professor with good English and experience in writing articles, from which he can make a publication of suitable quality.
Comments on the Quality of English LanguageExtensive editing of English language required. general style, scientific style etc
Author Response
This is my second time reviewing this manuscript. Compared to last time, there are some improvements in the quality of the material. In general, this article corresponds to the goals and objectives of the journal Polymers. However, it needs serious improvement. Firstly, the style of the text should be seriously revised until the impression remains that the article was written by students and was not checked by the professor. A significant improvement in the scientific style of the text is needed.
Response: We thank the reviewer for the time and suggestions to improve the quality of the manuscript. We sincerely apologize for the style of the manuscript. In the revised version we have extensively improved the style of the manuscript and the same was highlighted in red color.
Second, the manuscript requires proofreading by a native English speaker. A number of sentences are not written clearly. thoroughly checked the manuscript for English correction.
Response: We thank the reviewer for the corrections. The revised manuscript was thoroughly checked for English.
thirdly, a significant improvement is required in the quality of the text and illustrations (captions under the figures), the quality of the diagrams, many illustrations are auxiliary, which can be cited when defending a diploma or course work, but not in a scientific article (Figure 2, 4, etc.). All illustrations with three or four columns should be combined into one. illustrations should be done in a more specialized editor. Excel is not comme il faut.
Response: We thank the reviewer for the corrections. As per the suggestions, the mentioned figures were removed in the revised manuscript and the figures quality were improved.
It is completely unclear with which organisms the experiments were carried out; the authors isolated many strains, but the data are not clear on which ones are given. The title of the article contains fungi, but they also worked with bacteria, and the authors all call microbes. a table is needed with all the isolated strains and their identification, the strains need to be numbered and all results must be given with these numbers.
Response: We thank the reviewer for the suggestions. As per the comments, section 3.2 was revised with precise data.
The introduction resembles a student's thesis in style and materials. For example, the phrase We found very few literatures that deals with biodegradation with fungi. In my opinion, this indicates a problem with authors, but not a lack of literature.
Response: As per the suggestions, the introduction section has been revised properly.
The authors identified the strains; it is necessary to find more specific data on these strains in the literature and present them to formulate the goal of the work.
Response: In the revised manuscript more specific data has been included about the isolated strains.
Thus, I cannot recommend this manuscript for publication in this form, due to its very low quality (I have not listed half of the obvious problems), but it has potential for publication. It is worth finding a professor with good English and experience in writing articles, from which he can make a publication of suitable quality.
Response: We thank the reviewer for the suggestions. We have made significant corrections and modifications in the revised manuscript as per the suggestions.
Reviewer 2 Report
Comments and Suggestions for Authors
This work offers important new information about how microbial communities break down polyurethane (PU), emphasizing in particular the possible involvement of Aspergillus versicolor (ARF5) in this process. This work is interesting, which is a significant advancement over existing knowledge, but it needs substantial improvements before considering for publication. The publication is recommended, subjected to revision as mentioned below in comments to the authors.
· A thorough analysis of the body of research on microbial communities' degradation of petroleum underlays the introduction. Including a more thorough review would aid in putting the study's significance in perspective and point out any knowledge gaps.
Provide more information even though the techniques for identifying and verifying the degradation of PU-degrading microorganisms are mentioned. To improve the reproducibility and reliability of the results, a detailed methodology that includes detailed descriptions of the experimental procedures and controls should be provided.
· It is appropriate to confirm PU degradation using plate assays, Sturm tests, and scanning electron microscopy (SEM). Further characterization methods, such as gas chromatography-mass spectrometry (GC-MS) or Fourier-transform infrared spectroscopy (FTIR), might, nevertheless, offer more thorough insights into the degradation process.
Although the optimization of culture conditions for maximum PU degradation is mentioned, no information is given regarding particular parameters and how they affect degradation efficiency. The research would be strengthened if comprehensive optimization studies with conclusions and discussions were included.
· The one-month duration of the soil burial test may not be very long, even though it offers insight into degradation scenarios that occur in real life. To improve the robustness of the results, the duration could be extended or additional tests conducted in various environmental conditions could be conducted.
Although ARF5 isolation and identification are mentioned, the process for carrying out these actions is not explained. Clarity and repeatability of the results would be increased by including information on molecular identification procedures and isolation strategies.
· Although the optimization of the culture medium is mentioned, it is not made clear which specific parameters were optimized or how they affected the degradation of the PU. Incorporating comprehensive results and discussions on culture medium optimization would improve the study's comprehensiveness.
The Sturm test verifies the existence of CO2 as a byproduct of PU degradation; however, further verification by quantitative techniques may bolster the evidence for degradation.
· please support your statements in the discussion section with relevant references
· Also compare results with previous related study and show the advantages of this study
Author Response
This work offers important new information about how microbial communities break down polyurethane (PU), emphasizing in particular the possible involvement of Aspergillus versicolor (ARF5) in this process. This work is interesting, which is a significant advancement over existing knowledge, but it needs substantial improvements before considering for publication. The publication is recommended, subjected to revision as mentioned below in comments to the authors. A thorough analysis of the body of research on microbial communities' degradation of petroleum underlays the introduction. Including a more thorough review would aid in putting the study's significance in perspective and point out any knowledge gaps.
Response: We thank the reviewer for the suggestions. In the revised manuscript the introduction section has been thoroughly modified and corrected.
Provide more information even though the techniques for identifying and verifying the degradation of PU-degrading microorganisms are mentioned. To improve the reproducibility and reliability of the results, a detailed methodology that includes detailed descriptions of the experimental procedures and controls should be provided.
Response: We thank the reviewer for the corrections. The experimental section has been improved in the revised manuscript.
It is appropriate to confirm PU degradation using plate assays, Sturm tests, and scanning electron microscopy (SEM). Further characterization methods, such as gas chromatography-mass spectrometry (GC-MS) or Fourier-transform infrared spectroscopy (FTIR), might, nevertheless, offer more thorough insights into the degradation process.
Response: We thank the reviewer for the suggestions. The degradation was confirmed with FTIR and the same was mentioned in the manuscript.
Although the optimization of culture conditions for maximum PU degradation is mentioned, no information is given regarding particular parameters and how they affect degradation efficiency. The research would be strengthened if comprehensive optimization studies with conclusions and discussions were included.
Response: We thank the reviewer for the corrections. As per the comments the discussion section has been improved in the revised manuscript and the same was highlighted in red color.
Although ARF5 isolation and identification are mentioned, the process for carrying out these actions is not explained. Clarity and repeatability of the results would be increased by including information on molecular identification procedures and isolation strategies.Although the optimization of the culture medium is mentioned, it is not made clear which specific parameters were optimized or how they affected the degradation of the PU. Incorporating comprehensive results and discussions on culture medium optimization would improve the study's comprehensiveness.
Response: We thank the reviewer for the corrections. Extensive corrections has been made in the revised manuscript with more discussion and the process parameters affecting the PU degradation.
The Sturm test verifies the existence of CO2 as a byproduct of PU degradation; however, further verification by quantitative techniques may bolster the evidence for degradation. please support your statements in the discussion section with relevant references. Also compare results with previous related study and show the advantages of this study.
Response: We thank the reviewer for the suggestions. More explanations to support the results were included in the revised manuscript with relevant references.